# Chemical Constituents and Biological Activity Profiles on *Pleione* (Orchidaceae)

**DOI:** 10.3390/molecules24173195

**Published:** 2019-09-03

**Authors:** Xiao-Qian Wu, Wei Li, Jing-Xin Chen, Jun-Wen Zhai, Hui-You Xu, Lin Ni, Sha-Sha Wu

**Affiliations:** 1College of Landscape Architecture, Fujian Agriculture and Forestry University, Fuzhou 350002, China; 2Key Laboratory of National Forestry and Grassland Administration for Orchid Conservation and Utilization at College of Landscape Architecture, Fujian Agriculture and Forestry University, Fuzhou 350002, China; 3College of Plant Protection, Fujian Agriculture and Forest University, Fuzhou 350002, China

**Keywords:** *Pleione*, chemical structures, phenanthrenebibenzyl, bibenzyls, glucosyloxybenzyl succinate derivatives, anti-tumor activity, anti-neurodegenerative

## Abstract

*Pleione* (Orchidaceae) is not only famous for the ornamental value in Europe because of its special color, but also endemic in Southern Asia for its use in traditional medicine. A great deal of research about its secondary metabolites and biological activities has been done on only three of 30 species of *Pleione*. Up to now, 183 chemical compounds, such as phenanthrenes, bibenzyls, glucosyloxybenzyl succinate derivatives, flavonoids, lignans, terpenoids, etc., have been obtained from *Pleione.* These compounds have been demonstrated to play a significant role in anti-tumor, anti-neurodegenerative and anti-inflammatory biological activities and improve immunity. In order to further develop the drugs and utilize the plants, the chemical structural analysis and biological activities of *Pleione* are summarized in this review.

## 1. Introduction

Orchidaceae is one of the largest family of flowering plants. There are about 42 genus that are used for traditional medicine in China, but thus far, no phytochemical investigation has been conducted on 70% of them [1]. As one of unexplored medicinal orchid [2], *Pleione* contains about 30 species in the habitats of terrestrial, epiphytic or lithophytic, among which 12 are endangered [3]. It mainly distributes in China, Vietnam, Burma, Bangladesh and the Northeast Indian at elevation of 600–4200 m [4]. China is the central region, with 23 species distributed here and 12 of them are endemic [5,6,7,8].

The *Pleione* is attracting increasing attention nowadays in terms of the ornamental and medicinal values [9,10,11]. As a special colorful flower, *Pleione* was introduced to Europe from China in 1904 [4]. Currently, there are more than 400 cultivars [12]. In China, the dry pseudobulbs of *P. bulbocodioides* and *P. yunnanensis*, as well as *Cremastra appendiculata*, were the sources of the traditional Chinese medicine (TCM) ‘shan-ci-gu’. They are used for removing heat, counteracting toxicity, dissipating phlegm and resolving masses. They can also be applied on symptoms such as furuncles, carbuncles, scrofulous sputum, snake and insect bites, abdominal masses and lumps [13]. However, only three species of *Pleione*, including *P. humulis*, *P. praecox* and *P. maculate*, were used as traditional medicine in Northeastern India, applied on laceration wounds, colds, upper respiratory infection, liver complaints and stomach ailments [2]. In summary, five species have been commonly used in traditional medicines, but scientific studies have only been performed on three of them. More detailed research is needed to on *Pleione*.

The phytochemical research could be traced back to 1996. Li et al. [14,15,16,17,18,19,20] extracted 30 chemical compounds from the pseudobulbs of *P. bulbocodioides*, such as dihydrophenanthropyrans, bibenzyls, bichroman, polyphenol and flavan-3-ol. Among them, the dihydrophenanthropyran **35** was the first one isolated from *Pleione* and the chemical structure of dihydrophenanthropyran with a 4*H* [2, 1-*b*] pyran system was reported [15]. Although China is the *Pleione* distribution center, there was no phytochemical investigation there until 2007. Liu [1] initiated the chemical investigation of *P. bulbocodioides* leading to six novel compounds and 24 known compounds, among which 18 compounds were obtained from the *Pleione* for the first time. In addition, the ethyl acetate (EtOAc) extracted fraction from *P. bulbocodioides* was found to have a certain inhibitory effect on mice cancer cells LA795, exerting significant activity of anti-tumor. Since then, the EtOAc fraction has been the key research part. Due to few researches focused on *P. yunnanensis*, Dong et al. [21,22,23] carried out the phytochemical on *P. yunnanensis* in 2009, discovering 12 novel compounds and 12 known compounds. In addition, since the pseudobulbs of *P. formosana* have been used as the substitute of Shan-ci-gu in Taiwan Island, Shiao [24] performed chemical investigation on them in 2009 and isolated three novel compounds.

The structural diversity of *Pleione* prompted Wang’s group [25,26] to continue the phytochemical investigation and isolated sixty compounds from *P. bulbocodiodes* and *P. yunnanensis*, occupied more than 30% of the total compounds. It is noteworthy that all compounds were tested for several assays of biological activities in vitro, generally known as free radical scavenging activity, cytotoxic activity, inhibition of NO production activity and neurotoxicity activity. In addition, the experiment turned to the study on the high-polarity fraction, which led to the isolation of 10 glucosyloxybenzyl succinate derivatives [27]. Encouraged by previous phytochemical researches on *Pleione*, Li et al. [28,29] isolated pyrrolidone substituted bibenzyls and prenylated flavones from *P. bulbocodiodes*, which enriched the contents of the chemical constituents.

Moreover, there were 152 patents published referring to the *Pleione* in China; 88.8% of them were related to the pharmacological activity against breast cancer, lung cancer, liver cancer, stomach cancer, colon cancer, etc. In order to benefit future research on the phytochemistry and biological of the *Pleione*, this review will discuss about the chemical compound isolation from the *Pleione*, structural analysis of the metabolites and the corresponding biological activities evaluation.

## 2. Chemical Constituents

### 2.1. Phenanthrenes

Phenanthrenes is one of the typical compounds extracted from the *Pleione* [30]. Fifty-seven phenanthrenes have been isolated (Table 1 & Figure 1), including three simple dihydrophenanthrenes (**1**–**3**), 10 benzyl substituted dihydrophenanthrenes (**4**–**10**, **21**–**23**), three dihydrophenanthrene dimers (**11**–**13**), five dimers of phenanthrene (**15**–**17**, **24**, **25**), three phenanthrene and dihydrophenanthrene polymers (**14**, **18**–**19**), three dihydrophenanthrene and bibenzyl polymers (**20**, **26**, **27**), one dihydrophenanthrene and glycoside polymer (**28**), 18 dihydrophenanthrene and phenylpropanoid polymers (**29**–**46**), one phenanthrene and phenylpropanoid polymers (**47**) and 10 phenanthrene polymers (**48**–**57**). The spectroscopic data of Compound **1**–**57** are shown in Table 2 and Table 3. Of these compounds, the dihydrophenanthrene **1**, **3** and the bibenzyls **58** and **61** may be the main bioactive compounds for anti-tumor, anti-inflammatory and anti-oxidant activities, which might also be the result of their synergy [31]. The **27** was the first dihydrophenanthrene component connected with the bibenzyl isolated from the *Pleione* [15]. Three novel 9, 10-dihydrophenanthrofurans **29**–**31** were isolated by Dong in 2009 [21]. The properties of 9, 10-dihydrophenanthrofurans with a phenyl from the *Pleione* species were may be deemed to be a chemtaxonomic marker of this species. One dihydrophenanthrene connecting with a *β*-d-glucopyranosyl **28** as the first reported structure of glucosyloxybenzyl 2-isobutylmalates in *Pleione* was also obtained in 2013 [22]. The compounds synthesized by aldol reaction of condensation of acetone with 9, 10-phenanthrene were firstly obtained from *Pleione*
**48**, **49** by Wang [25] in 2014. Wang [26] demonstrated that there was a structure-biological activity relationship of phenanthrenes isolated from *P**. yunnanensis*. The structure of **39** is very similar with **40**, but the former showed stronger neurotoxic activity than the latter. This is because that the carbon-9′ (C9′) of **39** is acetylation, resulting in the carbonyl oxygen atom hydrogen to the receptor. Based on Wang’s research [25], Shao [32] determined eight phenanthrenequinone enantiomers configuration of **50**–**57** by means of the spectroscopy techniques, such as Nuclear Magnetic Resonance (NMR), High Resolution Electrospray Ionization Mass Spectroscopy (HRESIMS) and Executive Creative Director (ECD). The possible biosynthetic pathways can be inferred based on structural analysis.

### 2.2. Bibenzyls

Bibenzyls are also abundant in *Pleione* with a number of 44 (Table 4 & Figure 2) [30], including nine simple bibenzyls (**58**–**66**), 23 benzyl substituted bibenzyls (**67**–**89**), one bibenzyl and fluorene polymer (**85**), five bibenzyl and glycoside polymers (**91**–**95**), two bibenzyl and phenylpropanoid polymers (**96**, **97**) and four bibenzylamide polymers (**98**–**101**). The spectroscopic data of Compound **1**–**57** are shown in Table 5 and Table 6. The character of bibenzyls is that the carbon-3 (C3), carbon-5 (C5) and carbon-4′(C4′) positions are often hydroxyl or methoxy on the core structure, and the carbon-2 (C2) or/and carbon-4 (C4) often have a *p*-hydroxyl or phenyl substitution. Two typical structures were isolated from *Pleione* in 1997 [30]. One was a single structure that mediates an ether combined dihydrophenanthrene with dibenzyl **27**, the other was the bibenzyl with two *p*-hydroxybenzyl groups **76**, **77**. Another four bibenzyls **67**, **68**, **78**, **79** have only hydroxyl and *p*-hydroxybenzyl substituents, which is not common in bibenzyl derivatives [23]. It was meaningful to understand such particular structures. Li [28] isolated four pyrrolidone substituted bibenzyl **98**–**101** from *P. bulbocodiodes*, which further enriched the chemical compounds of the *Pleione* species.

### 2.3. Glucosyloxybenzyl Succinate Derivatives

The *Pleione* is also rich in glucosyloxybenzyl succinate derivatives [41]. The succinic acid is the basic structure and it often combines with saccharides to form glycosides (**102**–**124**) (Table 7 and Figure 3). The biological studies indicated that the glucosyloxybenzyl succinate derivatives compounds were documented to exert significant activities against delaying aging and improving learning and memory ability of aging mice [42]. Cui [43] used the succinic acid derivatives as an indicator compound for High Performance Liquid Chromatography (HPLC) content determination. The result indicated that the **117** and **118** can be used to distinguish three sources of TCM shan-ci-gu. Lv [44] established the HPLC fingerprint analysis of the *P. bulbocodioides* via measuring the content of the indicator compound **117**. The similar values in the ten producing areas were all more than 0.980 of Chinese medicine. The relative retention time (in the fingerprints was similar, but the Relative Standard Deviation (RSD) values of the relative peak areas were quite different. This was assumed to be the effects of the wild environment and growth years. On account of the difficulty to obtain high-polarity compounds **117**, **118**, **123** and **145**, Wang [45] developed a rapid and efficient method Elution-extrusion Counter-current Chromatography Separation (EECCC): the solvent system composed of *n*-butanol, ethanol and water with a volume ratio of 20:1:20. The upper phase was stationary phase, the lower phase was mobile phase at a flow rate of 1.5 mL·min^−1^, with a rotation speed of 850 rpm at temperature of 35 °C. Five high-polarity compounds were extracted in 371 min simultaneously by this method. The other four kinds of acids, the glucosyloxybenzyl succinate derivatives Pleionosides A–J (**102**–**111**) connected, are (2*R*)-2-*p*-hydroxybenzylmalic acid (**102**–**105**), (2*R*)-2-benzylmalic acid (**106**), (2*R*, 3*S*)-2-benzyl tartaric acid (**107**) and (2*R*)-2-isobutylmatic (**108**–**110**). Their properties confirmed that they could support further chemotaxonomic researches in Orchidaceae as the specialized metabolites [27].

### 2.4. Other Compounds

Other compounds consist of seven flavones (**125**–**131**), eight lignans (**132**–**139**) and 44 others (**140**–**183**) (Table 8 & Figure 4). The flavones contained three simple flavones (**125**–**127**), two prenylated flavones (**128**, **129**) and two biflavonoids (**130**, **131**). The lignans consist of three simple lignans (**132**–**134**) and five tetrahydrofuran lignans (**135**–**139**). Yuan [40] isolated biflavonoids **131** from *Pleione* for the first time in 2012. Li [16] isolated two isomerized lignan compounds **132** and **133** from *P. bulbocodioides* in 1997. The pseudobulbs of *P. formosana* have been used as one of the substitute of *Shan-ci-gu* [46,47]. However no phytochemical investigation was performed on it. Thus, Shiao [24] began the chemical research in 2009 and it was the first time to isolate the cycloartane triterpenoid compound **167** from the natural product. Yang [48] analyzed the chemical compounds of the *P. bulbocodiodes*, *P. yunnanensis* and *P. limprichtii* from fifteen producing areas by High Performance Liquid Chromatography Diode Array Detection (HPLC-DAD). The Cluster Analysis and Principal Component Analysis were used for quality evaluation, but the compounds corresponding to the chromatographic peak were not determined.

## 3. Biological Activities

Previous studies showed that the compounds extracted from *P. bulbocodiodes, P. yunnanensis* and *P. formosana* exerted anti-tumor, anti-neurodegenerative, anti-inflammatory anti-oxidation activities. That is why *Pleione* has been gaining increasing attention. The *Pleione*’s biological activities are tightly related to the traditional efficacy of “curing fever, detoxifying the body, mitigating the swelling and cleaning the blood stasis” in Chinese Pharmacopoeia [13]. Research on biological activities will establish a foundation for the further pharmacological researches and enlighten the drug discovery for anti-tumor usage.

### 3.1. Anti-Tumor Activity

The biological activities of *Pleione* can be attributed primarily to the phenanthrenes and bibenzyls. Among those activities, that against tumors was the most significant. Liu [37] proved that the ethyl acetate extract of *P. bulbocodiodes* had a certain inhibitory effect on mice cancer cells LA795, while the petroleum ether extract only had an inhibition rate of 75.58% at 800 μg·mL^−1^, but no remarkable inhibition at 400 μg·mL^−1^ and below. However, the *n*-butanol extract did not exert inhibitory at all. This result laid the foundation for the later chemical compounds study, and regarded the ethyl acetate as key fraction for research. The compounds such as **26**, **34**, **44**, **58**, **60** and **153** were demonstrated certain inhibitory effects against LA795 at 100 μg·mL^−1^. Liu [37] confirmed that **34** and **153** showed the cytotoxic activity against LA795 cells with IC_50_ value of 66 and 12 μg·mL^−1^. Compound **58** exhibited cytotoxic activity and anti-allergic activity. Wang [33] found that the bibenzyls **58** and **61** isolated from *P. bulbocodiodes* significantly inhibited the growth of leukemia cells K562, HL-60, liver cancer cells BEL-7402, gastric cancer cells SGC-7901, lung cancer cells A569 [50], H460 and melanoma cells M14. Wang’s group [32,34] indicated that **58** isolated from *P. yunnanensis*, performed strong activity against the growth of LA795 cells with IC_50_ value of 76.21 μM, but only moderate inhibition against A569 cells and BEL-7402 cells. Compound **40** was shown to exert moderate cytotoxic activity against A569 cells. Compound **48** was proved significant cytotoxic activity against cancer cells at 10^−6^ M. Compound **49** exerted cytotoxic activities against colon cancer cells HepG2, liver cancer cells BGC-823 and breast cancer cells MCF-7 with IC_50_ values of 8.3, 2.3 and 2.5 μM, respectively [32]. Compound **56** exerted moderate activities in colon cancer cells HCT-116, HepG2 cells and MCF-7 cells with IC_50_ values of 8.1, 8.4 and 3.9 μM, respectively. It suggested that the stereochemistry of 9(10)*H*-phenanthren-10(9)-one is of great significance to the cytotoxic activity. Tumor cell invasion and metastasis determined the prognosis of cancer patients [51]. In a word, a number of studies confirmed that the compounds isolated from the *Pleione* have an optimistic effect on anti-tumor treatment.

### 3.2. Anti-Neurodegenerative Activity

Glycosides were found to inhibit the proliferation of tumor cells, which is meaningful for anti-tumor therapy [52]. The glucosyloxybenzyls were subjected to evaluation for learning and memory deficits of mice caused via scopolamine and D-Gal + NaNO_2_ [53]. Zhang [54] discussed that the Dactylorhin B, **117**, **118** and **120** isolated from *Coeloglossum viride* var*. bractestum* were demonstrated to exert activities of anti-apoptosis, promoting intelligence and delaying aging. The *P. bulbocodiodes* consists of the three components mentioned above, except Dactylorhin B [27]. **145** was documented to exhibit activities of neuroprotective, neurasthenia and epilepsy [55]. **29** and **58** performed certain neurotoxic activities of mice hippocampal neurons (SY-SH-5Y) at 10^−5^ M [25]. In addition, **2**, **32** and **39** indicated significant neurotoxic activity at 10^−5^ M [45]. Han [27] reported the hepatoprotective activity of glucocopyloxybenzyl succinate derivatives for the first time in 2019. These neuroprotective effects may be related to the management of antioxidants, malondialdehyde (MDA), glutathione (GSH) levels as well as the improvement of adenosine triphosphatase (ATPase) [56]. The active metabolite of APAP was reported to deplete the glutathione and initiate mitochondrial oxidative stress. Moreover, the reactive oxygen species (ROS) produced during the latter process would destroy the normal function of the mitochondria, ultimately leading to the death of necrotic cell [57,58,59]. These studies fully clarified the pharmacodynamic basis of the *Pleione* and laid a material foundation for anti-dementia activity.

### 3.3. Anti-Inflammatory and Anti-Oxidation Activity

Some compounds of *Pleione* exert activities of anti-bacterial and anti-inflammatory (Table 9). Wang [25,33] illustrated that **1** significantly inhibited NO production in mice peritoneal macrophages at 10^−5^ M. Compounds **1** and **3** had strong inhibition activity on NO production. Compounds **3** and **61** showed good performance in calmodulin inhibition and antifungal action. Li [28,29] suggested that the compounds of **4**, **63** and **64** significantly inhibited NO production induced via LPS in BV-2 cells with IC_50_ values of 5.44, 2.46 and 3.14 μM, respectively. They may be the promising compounds for the development of anti-inflammatory drugs. However **9**, **25**, **26**, **58**, **62**, **73**, **74**, **75**, **81**, **82**, **86** and **117** only exhibited moderate inhibition on NO production. Liu [49] suggested that **170** was documented to have strong activities of anti-cytotoxicity and anti-bacterial.

### 3.4. Others

In addition to the anti-tumor, anti-neurodegenerative, anti-inflammatory and anti-oxidation activities, the *Pleione* also exhibited activities of inhibiting antigen-induced degranulation, free radical scavenging as well as anti-oxidant. Wang [26,34] proved that **58**, **69**, **70**, **76** and **81** were shown the activity of antigen-induced degranulation in RBL-2H3 cells [60]. The inhibition efficiency of **58**, **69** and **70** was between 65.5% and 99.4%.

## 4. Conclusions

The chemical investigation of the *Pleione* has attracted much attention around the world and some breakthrough progress has been made. Up to now, the family of the compounds had become more and more abundant, especially 9(10)*H*-phenanthren-10(9). This can not only provide significant evolutionary and chemotaxonomic knowledge of the genus *Pleione*, but also enlighten the further development and utilization of new drugs.

The future important focal points on the *Pleione* researches are summarized as follows. Firstly, the research range of species of the genuns *Pleione* need to be widened except for the *P. bulbocodioides*, *P. yunnanensis* and *P. formosana* in order to seek for novel substitutes. The mechanism of the biological activity should be figured out to shine more clear therapy pattern. Secondly, water-soluble and fat-soluble extracts are necessary to be explored. Thirdly, research needs further progress for clinical application to serve for the patients. Lastly, there needs to be immediate scientific protection for *Pleione* plants because of their endangered status. 

## Figures and Tables

**Figure 1 molecules-24-03195-f001:**
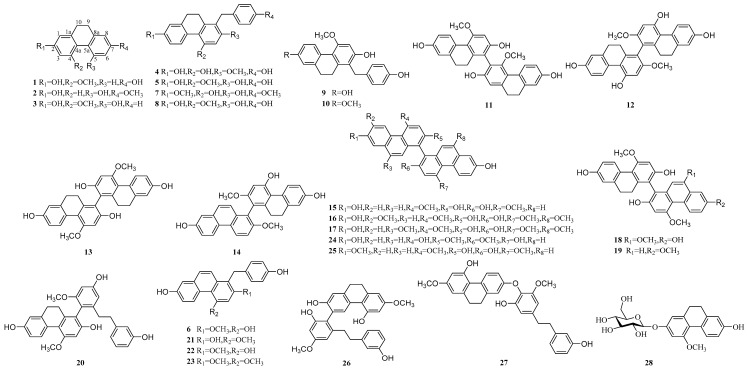
The structures of Phenanthrenes.

**Figure 2 molecules-24-03195-f002:**
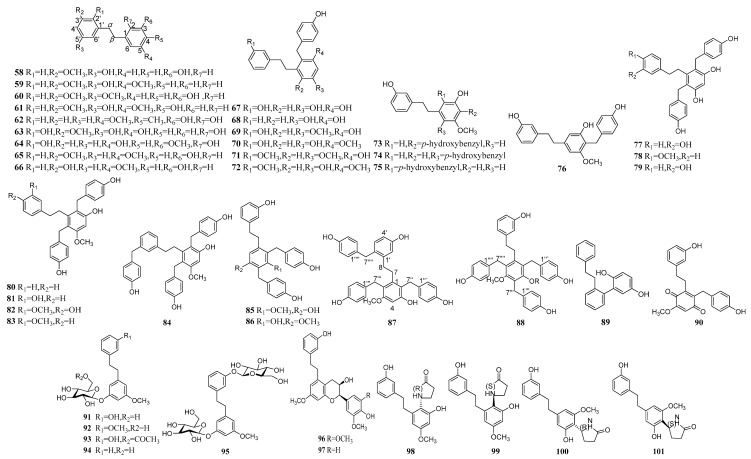
The structures of Bibenzyls.

**Figure 3 molecules-24-03195-f003:**
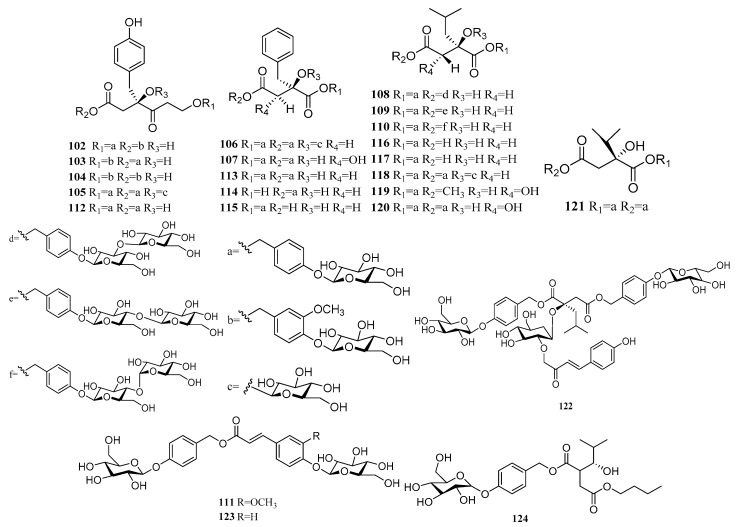
The structures of Glucosyloxybenzyl succinate derivatives.

**Figure 4 molecules-24-03195-f004:**
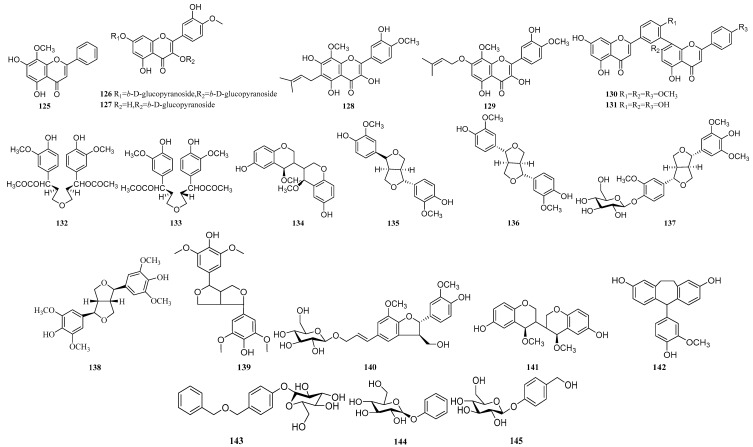
The structures of other compounds.

**Table 1 molecules-24-03195-t001:** Phenanthrenes from *Pleione* genus.

No.	Compound	Plant	Reference	No.	Compound	Plant	Reference
**1**	Coelonin	B *, Y *	[22,29]	**30**	Pleionesin B	Y	[33,34]
**2**	Lusianthridin	B, Y	[22,29]	**31**	Pleionesin C	Y	[33,34]
**3**	Hircinol	B	[33]	**32**	Shanciol H	B, Y	[21,35]
**4**	4, 7-dihydroxy-1-(*p*-hydroxybenzyl)-2-methoxy-9, 10-dihydrophenanthrene	Y	[22,34]	**33**	(4′-hydroxy-3′-methoxyphenyl)-10-hydroxymethyl-11-methoxy-5, 6, 9, 10-tetrahydrophenanthrene[2, 3-*b*]furan-3-ol	B	[36]
**5**	2, 7-dihydroxy-4-methoxy-1-(*p*-hydroxybenzyl)-9, 10-dihydrophenanthrene	Y	[22]	**34**	hydroxy-9-(4′-hydroxy-3′-methoxyphenyl)-11-methoxy-5, 6, 9, 10-tetrahydroohenanthrene-azaspiro[2, 3-*b*]furan-10-yl)methylethyl	B	[37]
**6**	1-(*p*-hydroxybenzyl)-4, 7-dihydroxy-2-methoxy-9, 10-dihydrophenanthrene	B	[6]	**35**	Shanciol	B, Y	[15]
**7**	Pleioanthrenin	F *	[24]	**36**	Shanciol E	B	[20]
**8**	2, 7-dihydroxy-1-(*p*-Hydroxybenzyl)-4-methoxy-9, 10-diphenanthrene	B, Y	[22,33]	**37**	(7′*S*, 8′*R*)-7-hydroxy-7-(4′-hydroxy-3′, 5′-dimethoxy-phenyl)-8′-hydroxymethyl-5-methoy-9, 10, 7′, 8-tetra hydro-phenanthrene-[2, 3-b]furan	B	[25]
**9**	(4-Hydroxybenzyl)-4-methoxy-9, 10-dihydrophenanthrene-2, 7-diol	B	[28]	**38**–**39**	Pleionesin D–E	Y	[25]
**10**	(4-hydroxybenzyl)-4, 7-dimethoxy-9, 10-dihydrophenanthrene-2-ol	B, F	[24,28]	**40**	(7′, 8′-trans)-7-hydroxy-10-methoxy-7′-(4′-hydroxy-3′-methoxyphenyl)-8′-hydoxymethyl-9, 10, 7′, 8′-tetrahydro-[2, 1-*b*]furan	Y	[26]
**11**	Blestrianol A	B	[25]	**41**–**43**	Bletilol A–C	B	[11,15]
**12**	4, 4′, 7, 7′-tetrahydroxy-2,2′-dimethoxy-9, 9′, 10, 10′-tetrahydro-1, 1′-biphenanthrene	B	[28]	**44**	Shanciol F	B, Y	[20,21]
**13**	Blestriarene A	B	[33,34]	**45**	Shanciols C	B	[19]
**14**	Blestriarene B	B	[21]	**46**	Shanciol G	B	[35]
**15**	Blestriarene C	Y	[22]	**47**	Shanciols D	B	[19]
**16**–**20**	Bulbocodioidin G–K	B	[25]	**48**	Bulbocodioidin A	B	[25]
**21**	1-(*p*-hydroxybenzyl)-2, 7-dihydroxy-4-methoxy-phenanthrene	B	[22]	**49**	Bulbocodioidin B	B	[25]
**22**	Shancidin	Y	[22]	**50**	(9*R*) bulbocodioidins A	B	[32]
**23**	7-hydroxy-2,4-dimethoxy-1-(*p*-hydroxybenzyl)-pHenanthrene	Y	[33]	**51**	(9*S*) Bulbocodioidins A	B	[32]
**24**	Monbarbatain A	B	[33]	**52**	(9*R*) Bulbocodioidins B	B	[32]
**25**	2, 7, 2′-didroxy-4, 4′, 7′-trimethoxy-1, 1′-biphenanthrene	B	[28]	**53**	(9*S*) Bulbocodioidins B	B	[32]
**26**	Phoyunnanin A	B	[28]	**54**	(9*R*) Bulbocodioidins C	B	[32]
**27**	Shancilin	B	[14]	**55**	(9*S*) Bulbocodioidins C	B	[32]
**28**	Shancigusins G	B, Y	[22]	**56**	(10*S*) Bulbocodioidins D	B	[32]
**29**	Pleionesin A	Y	[21]	**57**	(10*R*) Bulbocodioidins D	B	[32]

Note: * B: bulbocodioides; * Y: yunnanensis; * F: formosana. The same as below.

**Table 2 molecules-24-03195-t002:** ^13^C NMR data of compounds **1**–**57**.

No.	Solvent	Position													
		1	2	3	4	4a	4b	5	6	7	8	8a	9	10	10a
**1**	MeOH-*d*_4_	106.9	154.6	97.9	157.7	115.4	124.8	128.6	112.2	156.0	113.6	139.1	30.4	29.8	140.4
**2**	MeOH-*d*_4_	108.3	159.1	99.3	156.1	116.8	126.2	130.0	115.0	157.5	113.6	140.5	31.8	31.2	141.8
**3**	Acetone-*d*_6_	109.7	156.3	99.8	158.5	114.6	128.1	154.7	118.2	120.1	128.1	144.1	31.6	31.8	141.3
**4**	Acetone-*d*_6_	118.0	156.8	98.9	155.1	117.1	125.9	130.1	114.5	156.1	113.4	139.8	30.5	27.1	140.4
**5**	DMSO-*d*_6_	116.9	154.2	98.0	155.1	115.0	124.1	128.8	112.6	155.0	113.8	138.5	19.4	26.0	138.9
**6**	MeOH-*d*_4_	-	-	-	-	-	-	-	-	-	-	-	-	-	-
**7**	MeOH-*d*_4_	118.3	156.0	96.1	156.3	120.1	126.1	130.5	130.5	157.4	114.6	140.6	30.9	27.5	140.7
**8**	DMSO-*d*_6_	116.9	154.2	98.0	155.1	115.0	124.1	128.8	112.6	155.0	113.8	138.5	19.4	26.0	138.9
**9**	MeOH-*d*_4_	118.6	157.2	99.3	155.6	117.5	126.5	130.3	114.6	156.0	113.5	140.4	30.9	27.4	140.9
**10**	MeOH-*d*_4_	141.6	155.4	99.1	158.5	117.5	126.5	130.3	113.5	156.1	114.7	140.8	31.0	28.2	140.3
**11**	Acetone-*d*_6_	120.3	157.8	115.2	155.4	116.8	125.6	129.1	112.0	156.0	114.1	140.8	30.7	28.3	140.1
**12**	Acetone-*d*_6_	118.4	157.6	99.4	155.1	115.8	126.3	130.2	113.4	156.1	114.7	140.2	30.6	28.4	141.1
**13**	MeOH-*d*_4_	115.7	158.4	98.9	155.4	117.5	126.5	130.3	113.5	156.1	114.7	141.6	31.0	28.2	140.8
**14**	MeOH-*d*_4_	115.0	160.3	100.3	155.4	117.8	126.6	130.5	113.6	155.4	114.8	142.4	30.9	28.3	140.8
**15**	MeOH-*d*_4_	111.0	157.8	99.6	153.2	114.2	123.5	128.7	110.7	154.0	116.6	133.4	126.7	124.7	132.4
**16**	Acetone-*d*_6_	109.2	155.3	98.3	160.0	112.3	126.4	130.2	117.6	155.3	106.0	128.1	154.3	101.4	135.8
**17**	Acetone-*d*_6_	109.3	155.3	98.2	160.0	112.4	126.5	130.2	117.6	155.3	106.0	127.9	154.3	101.3	135.6
**18**	Acetone-*d*_6_	109.9	154.8	98.2	159.7	112.2	126.4	130.2	117.6	155.3	106.1	127.9	154.4	100.9	135.2
**19**	Acetone-*d*_6_	111.0	154.6	100.3	160.0	116.2	126.1	130.1	117.2	157.6	109.1	133.9	128.6	125.4	134.7
**20**	MeOH-*d*_4_	116.2	154.7	99.0	157.9	116.8	126.3	130.1	113.4	156.0	114.7	140.2	30.7	28.2	140.8
**21**	Acetone-*d*_6_	114.4	153.5	100.2	158.4	116.4	125.4	130.3	117.3	155.2	111.9	133.8	128.2	124.4	133.7
**22**	MeOH-*d*_4_	-	-	-	-	-	-	-	-	-	-	-	-	-	-
**23**	Acetone-*d*_6_	117.0	156.2	97.1	159.1	117.0	125.4	130.7	117.5	155.9	112.3	134.1	128.7	124.6	133.5
**24**	Acetone-*d*_6_	110.0	160.2	100.4	155.1	116.5	125.4	130.2	112.0	155.3	117.4	135.1	128.3	125.6	134.1
**25**	Chloroform-*d*	105.7	153.2	98.5	160.4	116.6	125.0	129.4	117.0	156.0	108.4	133.1	129.1	123.9	133.8
**26**	Acetone-*d*_6_	106.1	159.4	101.6	156.0	115.8	126.2	132.9	120.9	154.0	115.6	139.4	30.5	31.6	141.5
**27**	MeOH-*d*_3_	100.9	158.8	98.9	157.1	114.8	140.0	126.9	112.7	153.6	110.6	135.2	30.0	30.5	136.9
**28**	MeOH-*d*_4_	101.2	156.4	108.4	159.9	118.6	125.6	131.0	113.9	156.5	115.0	140.8	31.2	31.9	141.9
**29**	MeOH-*d*_4_	109.2	158.8	125.0	125.2	127.6	116.8	159.1	99.3	157.7	108.3	142.0	31.8	31.5	140.7
**30**	MeOH-*d*_4_	116.0	160.8	94.1	160.0	118.5	126.3	130.5	114.0	156.6	115.3	140.5	31.1	28.2	138.1
**31**	MeOH-*d*_4_	115.7	160.5	93.9	159.7	118.2	126.0	130.2	113.7	156.2	115.0	140.2	30.8	27.9	137.8
**32**	MeOH-*d*_4_	109.5	159.1	125.3	125.5	127.9	117.1	159.4	99.6	158.0	108.6	142.3	32.1	31.8	141.0
**33**	MeOH-*d*_4_	104.9	159.6	117.3	155.2	120.3	124.6	128.0	113.1	155.6	114.0	139.5	29.9	30.8	141.8
**34**	MeOH-*d*_4_	106.2	160.5	117.7	156.4	121.7	125.6	129.2	114.4	156.9	115.2	140.7	31.0	32.0	143.5
**35**	MeOH-*d*_3_	132.0	112.1	99.8	157.6	118.8	126.1	130.4	114.8	156.4	113.7	140.4	26.5	30.7	139.9
**36**	MeOH-*d*_3_	131.1	154.7	99.8	157.7	118.9	126.1	130.4	113.7	156.4	114.8	140.4	26.5	30.7	139.9
**37**	MeOH-*d*_4_	108.7	158.7	125.9	125.2	126.7	116.5	158.7	99.1	157.4	108.1	141.4	31.4.	31.1	139.4
**38**	Acetone-*d*_6_	117.2	158.8	93.5	160.2	116.7	137.0	129.9	113.5	156.0	114.9	139.7	30.4	27.4	136.4
**39**	MeOH-*d*_4_	116.0	160.8	94.2	160.0	118.5	126.3	130.5	114.0	156.6	115.3	140.5	31.1	28.2	138.1
**40**	MeOH-*d*_4_	116.7	160.5	93.9	159.4	118.0	126.2	130.2	113.7	156.2	115.0	140.2	30.9	27.9	137.5
**41**	MeOH-*d*_4_	133.0	153.6	93.0	159.3	114.4	116.9	129.3	113.0	158.5	114.3	136.7	26.9	29.7	134.9
**42**	MeOH-*d*_4_	125.8	153.6	92.9	159.2	114.3	116.8	129.2	114.3	158.4	114.5	136.6	26.8	29.8	133.8
**43**	MeOH-*d*_4_	107.4	146.8	98.3	157.7	116.8	123.8	124.1	125.9	154.7	114.4	133.1	30.4	30.6	133.1
**44**	MeOH-*d*_3_	118.1	159.5	94.0	160.6	116.9	137.6	130.2	113.8	156.2	115.0	140.3	28.0	30.9	135.8
**45**	MeOH-*d*_3_	118.2	159.5	93.9	160.6	116.8	137.7	130.2	113.8	156.3	115.1	140.3	28.0	30.9	136.3
**46**	MeOH-*d*_4_	118.2	160.1	94.3	159.4	118.3	126.1	130.8	113.7	156.3	115.0	140.3	30.9	28.0	137.8
**47**	MeOH-*d*_3_	108.4	159.0	99.5	159.4	125.9	136.5	125.5	134.7	157.7	109.0	142.0	31.6	31.9	117.0
**48**	MeOH-*d*_4_	106.5	158.7	106.6	159.0	118.7	122.3	130.9	115.4	157.7	112.9	141.8	79.8	205.0	133.2
**49**	MeOH-*d*_4_	105.9	159.2	100.5	160.0	112.1	128.9	130.9	121.9	157.1	113.5	132.3	204.7	80.0	143.4
**50**	MeOH-*d*_4_	121.8	157.3	105.1	156.3	118.2	122.8	131.2	115.3	157.3	112.3	141.2	80.0	206.5	132.4
**51**	MeOH-*d*_4_	121.8	157.3	105.1	156.3	118.2	122.8	131.2	115.3	157.3	112.3	141.2	80.0	206.5	132.4
**52**	MeOH-*d*_4_	126.5	155.2	130.0	155.7	123.4	122.6	130.4	116.1	158.1	112.4	141.5	80.0	205.6	130.2
**53**	MeOH-*d*_4_	126.5	155.2	130.0	155.7	123.4	122.6	130.4	116.1	158.1	112.4	141.5	80.0	205.6	130.2
**54**	MeOH-*d*_4_	122.1	157.0	105.7	156.5	119.7	122.9	131.0	114.9	157.0	113.0	141.6	82.5	207.6	131.9
**55**	MeOH-*d*_4_	122.1	157.0	105.7	156.5	119.7	122.9	131.0	114.9	157.0	113.0	141.6	82.5	207.6	131.9
**56**	MeOH-*d*_4_	119.3	157.2	99.8	158.4	113.6	130.9	131.5	121.9	156.9	112.4	132.6	205.7	55.7	139.2
**57**	MeOH-*d*_4_	119.3	157.2	99.8	158.4	113.6	130.9	131.5	121.9	156.9	112.4	132.6	205.7	55.7	139.2

**Table 3 molecules-24-03195-t003:** ^1^H NMR data of compounds **1**–**57**.

No.	Solvent	Position								
		1	3	4	5	6	7	8	9	10
**1**	MeOH-*d*_4_	6.29 d (2.5)	6.38 d (2.5)	-	7.99 d (9.0)	6.61 m	-	6.61 m	2.61 m	2.61 m
**2**	Acetone-*d*_6_	6.36 d (2.0)	6.44 d (2.0)	-	8.03 d (8.5)	-	-	-	2.62 s	2.62 s
**3**	chloroform-*d*	6.51 s	-	-	-	6.86 d (7.5)	7.15 dd (8.0, 7.5)	6.95 d (8.0)	2.69–2.70 m	2.65–2.67 m
**4**	Acetone-*d*_6_	-	6.61 s	-	8.01 d (9.0)	6.65 br d (9.0)	-	6.67 br s	2.60 m	2.52 m
**5**	Acetone-*d*_6_	-	6.60 s	-	8.00 d (9.0)	6.65 br d (9.0)	-	6.64 d (2.0)	2.57–2.60 m	2.50–2.53 m
**6**	chloroform-*d*	-	6.47, s	-	8.14 d, (8.6)	6.62 m	-	6.60, d, 3.0	2.13–2.61 m	2.13–2.61 m
**7**	MeOH-*d*_4_	-	6.64 s	-	8.00 d (8.0)	6.60 dd (8.0, 2.0)	-	6.62 d (2.0)	2.58 m	2.50 m
**8**	Acetone-*d*_6_	-	6.60 s	-	8.00 d (9.0)	6.65 br d (9.0)	-	6.64 d (2.0)	2.57–2.60 m	2.50–2.53 m
**9**	MeOH-*d*_4_	-	6.51 s	-	7.96 d (8.5)	6.60 dd (8.5, 2.5)	-	6.58 d (2.5)	2.45–2.57 m	2.45–2.57 m
**10**	MeOH-*d*_4_	-	6.57 s	-	8.04 d (8.0)	6.64 dd (8.0, 2.0)	-	6.60 d (2.0)	2.51–2.57 m	2.28–2.36 m
**11**	MeOH-*d*_4_	-	6.57 s	-	8.03 d (8.4)	6.62 dd (8.4, 2.4)	-	6.59 d (2.4)	2.44–2.46 m	2.53–2.55 m
**12**	Acetone-*d*_6_	-	6.58 s	-	8.24 d (8.5)	6.70 dd (8.5, 2.7)	-	6.67 d (2.7)	2.56 m	2.52 m
**13**	Acetone-*d*_6_	-	6.60 s	-	8.09 d (8.5)	6.69 dd (8.5, 3.0)	-	6.66 d (3.0)	2.53 m	2.33 m
**14**	MeOH-*d*_4_	-	-	-	8.09 d (8.8)	6.64 dd (2.8, 8.8)	-	6.56 d (2.8)	2.45 m	2.21 m
**15**	MeOH-*d*_4_	-	6.93 s	-	9.40 d (9.2)	7.03 dd (9.2, 2.8)	-	7.00 d (2.8)	7.23 d (9.2)	6.92 d (9.2)
**16**	Acetone-*d*_6_	-	6.87 s	-	9.50 d (9.6)	7.19 dd (9.6, 2.4)	-	7.61 d (2.4)	-	6.44 s
**17**	Acetone-*d*_6_	-	6.88 s	-	9.51 d (9.0)	7.20 dd (9.0, 3.0)	-	7.62 d (3.0)	-	6.51 s
**18**	Acetone-*d*_6_	-	6.81 s	-	9.47 d (9.6)	7.18 dd (9.6, 3.0)	-	7.65 d (3.0)	-	6.60 s
**19**	Acetone-*d*_6_	-	6.98 s	-	9.52 d (9.6)	7.20 dd (9.6, 3.0)	-	7.31 d (3.0)	-	7.25 d (9.0)
**20**	Acetone-*d*_6_	-	6.59 s	-	8.08 d (9.0)	6.67 dd (9.0, 2.4)	-	6.64 d (2.4)	2.51 m	2.30 m
**21**	Acetone-*d*_6_	-	6.99 s	-	9.43 d (9.5)	7.12 dd (9.5)	-	7.19 d (2.5)	7.53 d (9.5)	7.80 d (9.5)
**22**	chloroform-*d*	-	6.47, s	-	8.14 d (8.6)	6.62 m	-	6.60, d, 3.0	2.13–2.61 m	2.13–2.61 m
**23**	Acetone-*d*_6_	-	7.24 s	-	9.55 d (9.0)	7.24 dd (9.0, 2.0)	-	7.30 d (2.0)	7.64 d (9.5)	7.92 d (9.5)
**24**	Acetone-*d*_6_	-	7.02 s	-	9.51 d (8.4)	7.19 dd (8.4, 3.0)	-	7.18 d (3.0)	7.37 d (9.0)	7.03 d (9.0)
**25**	chloroform-*d*	-	7.05 s	-	9.57 d (10.0)	7.29 dd (10.0, 3.0)	-	7.18 d (3.0)	7.49 d (8.5)	7.11 d (8.5)
**26**	Acetone-*d*_6_	6.39 d (2.5)	6.41 d (2.5)	-	8.21 s	-	-	6.85 s	2.76 m	2.76 m
**27**	MeOH-*d*_3_	-	6.42 d, 2.3	-	7.92 d (8.6)	6.71, dd, 8.6, 2.6	-	6.74, d, 2.6	2.68–2.70, m	2.68–2.70 m
**28**	MeOH-*d*_4_	6.69 d (2.7)	6.45 d (2.7)	-	8.18 d (8.7)	6.60 dd (8.7, 2.7)	-	6.57 d (2.7)	2.58 m	2.63 m
**29**	MeOH-*d*_4_	6.61 s	-	8.00 s	-	6.35 d (2.0)	-	6.26 d (2.0)	2.57–2.62 m	2.57–2.62 m
**30**	Acetone-*d*_6_	-	6.59 s	-	8.04 d (8.4)	6.68 dd (8.4,3.0)	-	6.69 d (3.0)	2.59–2.77 m	2.59–2.77 m
**31**	MeOH-*d*_4_	-	-	-	-	-	-	-	-	-
**32**	chloroform-*d*	6.74 s	-	8.07 s	-	6.42 d (2.0)	-	6.36 d (2.0)	2.70–2.71 m	2.70–2.71 m
**33**	MeOH-*d*_4_	6.54 s	-	-	8.00 d (9.0)	6.65 m	-	6.67 d (2.5)	2.68 m	2.68 m
**34**	MeOH-*d*_4_	6.57 s	-	-	8.03 d (9.6)	6.70 dd (9.6, 2.4)	-	6.68 d (2.4)	2.70 m	2.70 m
**35**	MeOH-*d*_3_	-	6.5, s	-	7.99 d (8.5)	6.62 dd (8.5, 2.5)	-	6.64, d, 2.5	2.62, m	2.62, m
**36**	MeOH-*d*_3_	-	6.51 s	-	8.00 d (8.5)	6.62 dd(8.5, 2.6)	-	6.65 d (2.6)	2.60–2.67 m	2.60–2.67 m
**37**	Acetone-*d*_6_	6.68 s	-	-	8.09 s	6.45 d (2.4)	-	6.38 d (2.4)	2.65 m	2.66 m
**38**	Acetone-*d*_6_	-	6.55 s	-	8.03 d (9.0)	6.67 dd (9.0, 2.5)	-	6.67 d (2.5)	2.55–2.73 m	2.55–2.73 m
**39**	Acetone-*d*_6_	-	6.53 s	-	8.09 d (8.5)	6.72 dd (8.5, 2.5)	-	6.69 d (2.5)	-	-
**40**	Acetone-*d*_6_	-	6.54 s	-	8.03 d (9.5)	6.67 dd (9.5, 2.5)	-	6.68 br s	2.63 m	2.63 m
**41**	MeOH-*d*_4_	-	6.53 s	-	8.09 d (8.6)	6.72 dd (8.6, 3.0)	-	6.69 d (3.0)	2.7 m	2.7 m
**42**	MeOH-*d*_4_	-	6.51 s	-	8.09 d (8.4)	6.71 dd (8.4, 2.6)	-	6.69 d (2.6)	2.67–2.69 m	2.67–2.69 m
**43**	MeOH-*d*_4_	6.34 d (2.1)	6.42 d (2.1)	-	8.08 s	-	-	6.74 s	2.67–2.76 m	2.67–2.76 m
**44**	MeOH-*d*_3_	-	6.54 s	-	7.99 d (9.4)	6.62 m	-	6.61 d (2.6)	2.56–2.70 m	2.56–2.70 m
**45**	MeOH-*d*_3_	-	6.56 s	-	8.00 d (9.2)	6.62 dd (9.6, 2.8)	-	6.61 d (2.8)	2.59–2.71 m	2.59–2.71 m
**46**	MeOH-*d*_4_	-	6.58 s	-	8.09 d (8.5)	6.63 m	-	6.65 d (2.5)	2.67 m	2.67 m
**47**	MeOH-*d*_3_	6.31 d (2.1)	6.41 d (2.1)	-	8.06 s	-	-	6.69 s	2.62–2.69 m	2.62–2.69 m
**48**	MeOH-*d*_4_	6.80 d (2.4)	6.76 d (2.4)	-	8.23 d (9.0)	6.69 dd (9.0, 2.4)	-	7.13 d (2.4)	-	-
**49**	MeOH-*d*_4_	6.83 d (2.4)	6.47 d (2.4)	-	8.30 d (8.4)	7.00 dd (8.4, 3.0)	-	7.13 d (3.0)	-	-
**50**	MeOH-*d*_4_	-	6.76 s	-	8.07 d (9.0)	6.67 dd (9.0, 2.4)	-	7.05 d (2.4)	-	-
**51**	MeOH-*d*_4_	-	6.76 s	-	8.07 d (9.0)	6.67 dd (9.0, 2.4)	-	7.05 d (2.4)	-	-
**52**	MeOH-*d*_4_	-	-	-	8.11 d (8.4)	6.72 dd (8.4, 2.4)	-	7.06 d (2.0)	-	-
**53**	MeOH-*d*_4_	-	-	-	8.11 d (8.4)	6.72 dd (8.4, 2.4)	-	7.06 d (2.0)	-	-
**54**	MeOH-*d*_4_	-	6.83 s	-	8.10 d (8.4)	6.81 d (2.4)	-	-	-	-
**55**	MeOH-*d*_4_	-	6.83 s	-	8.10 d (8.4)	6.81 d (2.4)	-	-	-	-
**56**	MeOH-*d*_4_	-	6.60 s	-	8.37 d (9.0)	7.02 dd (9.0, 2.4)	-	7.01 d (2.4)	-	3.86 d (9.6)
**57**	MeOH-*d*_4_	-	6.60 s	-	8.37 d (9.0)	7.02 dd (9.0, 2.4)	-	7.01 d (2.4)	-	3.86 d (9.6)

**Table 4 molecules-24-03195-t004:** Bibenzyls from *Pleione* genus.

No.	Compound	Plant	Reference	No.	Compound	Plant	Reference
**58**	Batatasin III	B, Y	[17,28]	**78**	Shancigusin A	Y	[23]
**59**	3′-*O*-methylbatatasin III	B, Y	[17,22]	**79**	Shancigusin B	Y	[23]
**60**	3, 5-Dimethoxy-3′-hydroxybibenzyl https://scifinder.cas.org/scifinder/view/text/javascript:;	B	[22]	**80**	Arundin	F	[24]
**61**	Gigantol	B	[33]	**81**	5-*O*-Methylshanciguol	B, Y, F	[23,24,28]
**62**	Bauhinol C	B	[28]	**82**	Blestritin B	B	[28]
**63**	2, 5, 2′, 5′-Tetrahydroxy-3-methoxybibenzyl	B	[28,38]	**83**	2, 6-bis-(4-hydroxybenzyl)-3′, 5-dimethoxy-3-hydroxybibenzyl	F	[24]
**64**	2, 5, 2′, 3′-tetrahydroxy-3-methoxybibenzyl	B	[28]	**84**	Bulbocodin	B	[18]
**65**	hydroxy-3′,5-dimethxoybibenzyl	Y	[22]	**85**	Bulbocodin C	B, F	[20]
**66**	3, 3′-dihydroxy-5-methoxybibenzyl	Y	[22]	**86**	Bulbocodin D	B	[20,31]
**67**	Shancigusin C	Y	[23]	**87**	Pleiobibenzynin A	F	[24]
**68**	Shancigusin D	Y	[23]	**88**	Pleiobibenzynin B	F	[24]
**69**	3, 3′-dihydroxy-2-(*p*-hydroxybenzyl)-5-methoxybibenzyl	B, Y, F	[18,23,38]	**89**	6′-(3′′-hydroxyphenethyl)-4′-methoxydiphenl-2, 2′, 5′-triol	B	[39]
**70**	3′, 5-dihydroxy-2-(*p*-hydroxybenzyl)-3-methoxybibenzyl	B, Y, F	[18,23,24]	**90**	2-(4′′-hydroxybenzyl)-3-(3′-hydroxy-phenethyl)-5-methoxy-cyclohexa-2, 5-diene-1, 4-dione	B	[39]
**71**	Gymconopin D	B	[36,37,40]	**91**	Batatsin III-3-*O*-glucoside	B, Y	[17,22]
**72**	Bulbocol	B, F	[18,24]	**92**	3′, 5-dimethoxybibenzyl-3-*O*-*β*-d-glucopyranoside	B, Y	[17,22]
**73**	Arundinin	Y, F	[28]	**93**–**94**	Shancigusins E-F	Y	[22]
**74**	Isoarundinin I	B	[28]	**95**	5-methoxyl bibenzyl-3, 3′-di-*O*-*β*-d-glucopyranoside	B	[27]
**75**	Isoarundinin II	B	[28]	**96**–**97**	Shanciols A–B	B	[19]
**76**	3, 3′-dihydroxy-4-(*p*-hydroxybenzyl)-5-methoxybibenzyl	B, Y	[18]	**98**–**101**	Dusuanlansins A–D	B	[28]
**77**	Shanciguol	B, Y	[14,23]				

**Table 5 molecules-24-03195-t005:** ^13^C NMR data of compounds **58**–**101**.

No.	Solvent	Position													
		1	2	3	4	5	6	1′	2′	3′	4′	5′	6′	C-α	C-β
**58**	Acetone-*d*_6_	145.1	108.8	159.3	99.7	161.8	106.2	144.3	116.2	158.2	113.6	130.0	120.4	38.6	38.2
**59**	Chloroform-*d*	146.5	107.7	156.4	98.8	161.3	106.4	145.1	113.7	159.9	111.5	129.5	120.4	35.8	36.0
**60**	Chloroform-*d*	144.1	106.6	160.7	97.9	160.7	106.6	143.7	115.4	155.6	129.5	120.9	112.9	38.0	37.5
**61**	Acetone-*d*_6_	145.1	108.9	159.2	99.7	161.8	106.3	134.1	115.5	148.0	145.2	112.9	121.6	38.0	39.1
**62**	Chloroform-*d*	141.8	128.5	128.3	125.9	128.3	128.5	140.7	103.6	154.0	110.0	158.7	108.1	37.8	37.9
**63**	MeOH-*d*_4_	116.9	142.0	159.2	99.4	157.6	108.5	126.1	140.6	113.7	130.2	156.2	115.1	31.9	31.4
**65**	Chloroform-*d*	144.0	107.8	158.0	98.8	159.8	105.4	143.2	113.9	160.9	128.8	120.6	111.1	37.6	37.4
**67**	MeOH-*d*_3_	144.0	118.6	157.5	101.4	157.0	108.0	145.0	116.2	158.3	113.7	130.2	120.7	36.5	38.6
**68**	MeOH-*d*_3_	143.9	118.6	157.5	101.3	157.1	108.8	143.4	129.4	129.2	126.7	129.2	129.4	36.6	38.6
**69**	Acetone-*d*_6_	143.6	119.1	157.0	100.1	159.6	107.0	144.5	113.6	158.2	116.1	130.0	120.3	36.2	38.1
**70**	Acetone-*d*_6_	143.2	119.3	159.7	97.8	157.5	109.1	144.5	113.6	158.3	116.1	130.1	120.3	36.0	38.2
**71**	Chloroform-*d*	142.4	119.9	155.5	96.5	158.8	105.8	143.8	115.3	158.9	112.8	129.5	120.8	35.2	37.2
**72**	MeOH-*d*_3_	143.7	121.8	161.2	98.1	157.6	109.5	144.9	114.9	160.3	112.6	130.2	120.0	36.4	38.6
**73**	MeOH-*d*_4_	-	-	-	-	-	-	-	-	-	-	-	-	-	-
**74**	Chloroform-*d*	142.6	122.2	158.6	106.4	150.8	113.2	142.9	121.3	150.4	119.1	129.1	125.7	36.8	34.8
**75**	Chloroform-*d*	142.1	124.3	150.8	102.7	158.4	114.2	143.0	121.3	150.0	119.0	129.1	125.6	36.7	34.5
**76**	Acetone-*d*_6_	141.9	109.9	156.3	114.9	159.3	103.5	144.4	115.4	158.3	113.6	129.9	120.3	38.6	38.4
**77**	MeOH-*d*_3_	145.4	119.1	155.7	101.8	155.7	119.1	142.8	116.1	168.3	113.8	130.2	120.6	33.4	37.7
**78**	MeOH-*d*_3_	142.9	118.9	155.6	101.5	155.6	118.9	134.7	130.1	116.0	156.3	116.0	130.1	33.8	36.9
**79**	MeOH-*d*_3_	142.7	118.9	155.6	101.6	155.6	118.9	143.8	129.3	129.2	126.7	129.2	129.3	33.5	37.7
**80**	MeOH-*d*_4_	142.6	119.5	158.4	98.0	155.9	120.2	143.6	129.1	129.2	126.7	129.2	129.1	33.4	37.7
**81**	Acetone-*d*_6_	142.2	119.9	157.8	98.0	158.3	119.0	144.7	113.6	156.0	115.7	130.1	120.1	33.2	37.2
**82**	MeOH-*d*_4_	143.2	121.8	158.7	98.4	156.2	120.7	135.6	113.3	149.0	145.8	116.3	121.8	34.0	38.2
**83**	MeOH-*d*_4_	142.7	120.3	156.0	98.1	161.0	119.5	145.2	112.6	158.4	114.5	130.2	121.5	37.6	33.3
**84**	MeOH-*d*_3_	142.9	120.4	156.7	98.4	158.5	119.5	131.5	155.9	113.9	116.6	142.8	132.1	34.7	37.9
**85**	MeOH-*d*_3_	144.9	124.4	159.5	120.5	158.3	113.6	141.9	116.4	156.2	113.8	130.2	120.8	29.8	31.6
**86**	MeOH-*d*_4_	145.0	130.6	157.9	120.7	158.3	106.0	140.9	116.5	154.8	113.8	130.2	120.9	29.0	31.4
**87**	MeOH-*d*_4_	140.8	119.4	156.3	98.2	158.4	120.2	142.7	116.5	156.6	113.8	132.0	131.4	31.7	34.6
**88**	MeOH-*d*_4_	140.8	124.9	154.1	121.1	157.7	125.7	145.1	116.0	158.3	113.7	130.1	120.6	33.4	37.7
**89**	MeOH-*d*_4_	143.1	117.6	155.6	99.1	160.0	106.3	144.0	115.0	157.0	112.4	128.9	119.6	36.6	37.1
**90**	MeOH-*d*_4_	143.4	145.3	189.1	108.0	160.1	184.0	144.0	116.3	158.5	114.1	130.4	120.7	30.2	35.6
**91**	MeOH-*d*_4_	144.6	110.6	158.4	102.7	162.2	109.7	145.6	116.6	160.2	114.0	130.3	121.0	38.6	38.6
**92**	MeOH-*d*_4_	144.6	110.5	160.2	102.6	162.1	109.7	145.4	115.4	161.2	112.5	130.3	122.1	38.8	38.8
**93**	MeOH-*d*_4_	144.5	110.4	158.4	102.0	162.2	109.2	145.4	116.5	160.0	113.9	130.3	120.9	38.7	39.1
**94**	MeOH-*d*_4_	145.4	110.3	160.1	101.6	162.0	109.5	143.0	129.6	129.3	126.9	129.3	129.6	39.2	38.7
**95**	DMSO-*d*_6_	143.8	108.9	158.6	99.9	160.2	107.8	143.1	116.5	157.5	113.6	129.2	122.0	37.0	36.7
**96**	Chloroform-*d*	143.3	109.5	160.4	100.5	156.6	112.5	144.6	116.5	158.5	113.9	130.4	120.9	35.9	37.8
**97**	Chloroform-*d*	143.3	109.4	160.4	100.5	156.6	112.5	144.6	116.1	158.5	114.0	130.4	121.4	35.9	37.8
**98**	MeOH-*d*_4_	143.4	119.5	159.0	101.4	161.1	107.4	144.4	120.9	158.5	114.0	130.3	116.5	36.8	39.7
**99**	MeOH-*d*_4_	143.4	119.5	159.0	101.4	161.1	107.4	144.4	120.9	158.5	114.0	130.3	116.5	36.8	39.7
**100**	MeOH-*d*_4_	144.4	109.8	157.4	114.8	160.2	104.3	144.5	116.4	158.4	113.8	130.2	120.8	39.0	38.7
**101**	MeOH-*d*_4_	144.4	109.8	157.4	114.8	160.2	104.3	144.5	116.4	158.4	113.8	130.2	120.8	39.0	38.7

**Table 6 molecules-24-03195-t006:** ^1^H NMR data of compounds **58**–**101**.

No.	Solvent	Position											
		2	3	4	5	6	2′	3′	4′	5′	6′	H-*α*	H-*β*
**58**	Acetone-*d*_6_	6.30 dd (1.8, 1.8)	-	-	-	6.32 dd (1.8, 1.8)	6.71 d (1.8)	-	6.65 dd (7.8, 1.8)	7.07 dd (7.8, 7.8)	6.70 d (7.8)	2.75–2.80 m	2.75–2.80 m
**59**	Chloroform-*d*	6.40 t (2.0)	-	6.46 t (2.0)	-	6.23 t (2.0)	-	-	6.72 m	7.08 t (9.0)	6.72 m	-	-
**60**	Chloroform-*d*	6.33 m	-	6.31 t (2.5)	-	6.33 m	6.66 m	-	6.66 m	7.15 t (7.5)	6.76 m	2.85 m	2.85 m
**61**	Acetone-*d*_6_	-	-	-	-	6.29 dd (1.8, 1.8)	6.80 d (1.8)	-	-	6.71 d (7.8)	6.65 dd (7.8, 1.8)	2.27–2.59 m	2.27–2.59 m
**62**	Chloroform-*d*	7.24 m	7.33 m	7.33 m	7.33 m	7.24 m	6.32 br s	-	-	-	6.36 br s	-	-
**63**	MeOH-*d*_4_	-	-	6.39 d (2.1)	-	6.30 d (2.1)	-	6.62 br s	7.99 d (8.8)	-	6.60 d (2.6)	2.63 br s	2.63 br s
**65**	Chloroform-*d*	-	-	6.17 t (1.5)	-	6.23 m	6.71 m	-	6.72 m	7.15 t (8.0)	6.75 m	2.79 m	2.79 m
**67**	MeOH-*d*_3_	-	-	6.23 d (2.0)	-	6.19 d (2.0)	6.55 br s	-	6.57 br d (8.0)	7.03 t (8.0)	6.54 br d (8.0)	2.65–2.69 m	2.52–2.55 m
**68**	MeOH-*d*_3_	-	-	6.17 d (2.4)	-	6.13 d (2.4)	6.98 d (7.2)	7.14 t (7.2)	7.07 t (7.2)	7.14 t (7.2)	6.98 d (7.2)	2.61–2.64 m	2.51–2.55 m
**69**	Acetone-*d*_6_	-	-	6.39 d (1.5)	-	6.36 d (1.5)	6.67 br s	-	6.62 br d (8.0)	7.05 t (8.0)	6.62 br d (8.0)	2.77 m	2.61 m
**70**	Acetone-*d*_6_	-	-	6.39 d (2.0)	-	6.37 d (2.0)	6.65 br s	-	6.61 br d (8.0)	7.05 t (8.0)	6.63 br d (8.0)	2.73 m	2.59 m
**71**	Chloroform-*d*	-	-	6.34 d (2.0)	-	6.29 d (2.0)	6.49 br s	-	6.69 m	7.11 t (8.5)	6.64 m	2.79 m	2.65 m
**72**	MeOH-*d*_3_	-	-	6.34 d (2.6)	-	6.27 d (2.6)	6.55 t (2.1)	-	6.69 dd (7.9, 1.7)	7.11 t (7.9)	6.64 m	2.56–2.64 m	2.56–2.64 m
**73**	MeOH-*d*_4_	6.19 d (2.0)				6.25 d (2.0)	6.55 s		6.52 m	6.99 t (8.0)	6.59 dd (8.0, 2.0)	2.71–2.74 m	2.71–2.74 m
**74**	Chloroform-*d*	-	-	6.67 d (2.1)	-	6.79 d (2.1)	6.90 d (3.0)	-	6.93 d (8.4)	7.25 t (8.4)	7.16 d (8.4)	2.70 m	2.82 m
**75**	Chloroform-*d*	-	-	6.61 d (2.1)	-	6.58 d (2.1)	6.82 br	-	6.97 d (8.2)	7.26 apt t (8.7, 8.1)	6.97 dd	2.73 m	2.85 m
**76**	Acetone-*d*_6_	6.37 s	-	-	-	6.40 s	-	-	6.67 br d (7.2)	7.05 t (7.8)	6.70 br s (8.0)	2.86 m	2.75 m
**77**	MeOH-*d*_3_	-	-	6.39 s	-	-	6.51 t (2.1)	-	6.56 ddd (7.7, 2.6, 1.9)	7.01 t (7.7)	6.47 d (7.7)	2.24–2.30, m	2.24–2.30, m
**78**	MeOH-*d*_3_	-	-	6.32 s	-	-	6.74 d (8.4)	6.57 d (8.4)	-	6.57 d (8.4)	6.74 d (8.4)	2.54–2.57 m	2.16–2.20 m
**79**	MeOH-*d*_3_	-	-	6.34 s	-	-	6.92 d (7.2)	7.13 t (7.2)	7.04 t (7.2)	7.13 t (7.2)	6.92 d (7.2)	2.58–2.62 m	2.24–2.28 m
**80**	MeOH-*d*_4_	-	-	-	-	-	6.97 d (7.0)	7.18 t (7.5)	7.10 t (7.5)	7.18 t (7.5)	6.97 d (7.0)	2.66–2.72 m	2.66–2.72 m
**81**	Acetone-*d*_6_	-	-	6.58 s	-	-	6.64 br s	-	6.61 m	7.04 t (8.0)	6.58 m	2.74 m	2.37 m
**82**	MeOH-*d*_4_	-	-	6.48 s	-	-	6.38 d (1.8)	-	-	6.60–6.65 m	6.44 dd (8.0, 1.8)	2.61–2.71 m	2.30–2.40 m
**83**	MeOH-*d*_4_	-	-	6.49–7.15 m	-	-	6.49–7.15 m	-	-	6.49–7.15	-	2.23–2.37 m	2.64–2.73 m
**84**	MeOH-*d*_3_	-	-	6.45 s	-	-	-	6.57 d (8.2)	6.53 dd (8.2, 3.3)	-	6.58 d (3.3)	2.43–2.46 m	2.43–2.46 m
**85**	MeOH-*d*_3_	-	-		-	6.56 s	6.50 s		6.57 dd (8.4, 2.2)	7.01 t (8.4)	6.50 m	2.59–2.66 m	2.59–2.66 m
**86**	MeOH-*d*_4_	-	-	-	-	6.33 s	6.53 m	-	6.57 dd (8.2, 2.3)	7.02 t (8.2)	6.53 m	2.59–2.77 m	2.59–2.77 m
**87**	MeOH-*d*_4_	-	-	6.46 s	-	-	6.57 d (2.0)	-	6.51 dd (8.5, 2.0)	6.77 *d* (8.5)	-	2.66 m	2.42 m
**88**	MeOH-*d*_4_	-	-	-	-	-	6.47 dd (2.1, 2.0)	-	6.55 ddd (8.0, 2.1, 2.0)	7.00 t (8.0)	6.46 ddd (8.0, 2.1, 2.0)	2.67 m	2.35 m
**89**	MeOH-*d*_4_	-	-	6.33 m	-	6.34 m	6.41 m	-	6.52 m	6.96 t (8.0)	6.39 m	2.56 m	2.56 m
**90**	MeOH-*d*_4_	-	-	6.02 s	-		6.59 m	-	6.62 m	7.06 t (8.0)	6.58 m	2.75 m	2.46 m
**91**	MeOH-*d*_4_	6.52 t (2.0)	-	6.51 t (2.0)	-	6.40 t (2.0)	6.59 t (2.0)	-	6.61 dd (7.5, 2.8)	7.05 t (7.7)	6.63 br d (7.7)	2.83 m	2.83 m
**92**	MeOH-*d*_4_	6.52 t (2.0)	-	6.50 t (2.0)	-	6.40 t (2.0)	6.70 br d (1.7)	-	6.73 dd (8.6, 7.7)	7.14 t (8.6)	6.74 br d (7.7)	2.84 m	2.84 m
**93**	MeOH-*d*_4_	6.41 m	-	6.40 m	-	6.35 br s	6.54 m	-	6.54 m	7.00 br t (7.8)	6.57 d (7.8)	2.71–2.81	2.71–2.81
**94**	MeOH-*d*_4_	6.51 m	-	6.51 m	-	6.39 s	7.15 m	7.24 t (7.5)	7.15 m	7.24 t (7.5)	7.15 m	2.83–2.91	2.83–2.91
**95**	DMSO-*d*_6_	6.51 br s	-	6.45 br s	-	6.44 br s	6.92 br s	-	6.86 br d (7.8)	7.18 t (7.8)	6.87 br d (7.8)	2.81 m	2.81 m
**96**	Chloroform-*d*	6.37 d (2.6)	-	6.31 d (2.6)	-	-	6.61 m	-	6.61 m	7.05 t (7.7)	6.63 d (7.7)	2.80 m	2.80 m
**97**	Chloroform-*d*	6.36 d (2.6)	-	6.29 d (2.6)	-	-	6.61 m	-	6.61 m	7.05 t (7.7)	6.64 d (7.7)	2.79 m	2.79 m
**98**	MeOH-*d*_4_	-	-	6.25 d (2.5)	-	6.22 d (2.5)	6.60 br s	-	6.61 m	7.07 t (7.5)	6.62 d (7.5)	2.94 m	2.76 m
**99**	MeOH-*d*_4_	-	-	6.25 d (2.5)	-	6.22 d (2.5)	6.60 br s	-	6.61 m	7.07 t (7.5)	6.62 d (7.5)	2.94 m	2.76 m
**100**	MeOH-*d*_4_	6.28 br s	-	-	-	6.27 br s	6.59 d (2.5)	-	6.57 m	7.07 t (7.5)	6.64 d (7.5)	2.76–2.81 m	2.76–2.81 m
**101**	MeOH-*d*_4_	6.28 br s	-	-	-	6.27 br s	6.59 d (2.5)	-	6.57 m	7.07 t (7.5)	6.64 d (7.5)	2.76–2.81 m	2.76–2.81 m

**Table 7 molecules-24-03195-t007:** Glucosyloxybenzyl succinate derivatives from Pleione genus.

No.	Compound	Plant	Reference	No.	Compound	Plant	Reference
**102**–**111**	Pleionosides A–J	B	[27]	**118**	Dactylorhin A	B, Y	[22,27]
**112**	Vandateroside II	B	[27]	**119**	(−)-(2*R*,3*R*)-1-(4-*O*-*β*-d-glucopyranosyloxybenzyl)-4-methyl-2-isobutyltartrate	B	[27]
**113**	Grammatophylloside B	B	[27]	**120**	Loroglossin	B	[27]
**114**	Grammatophylloside A	B	[27]	**121**	(−)-(2*S*)-1-[(4-*O*-*β*-d-glucopyranosyloxy) benzyl]-2-isopropyl-4-[(4-*O*-*β*-d-glucopyranosyloxy)benzyl] malate	B	[27]
**115**	Cronupapine	B	[27]	**122**–**123**	Shancigusins H-I	Y	[22]
**116**	Gymnoside I	B, Y	[22,27]	**124**	Bletillin A	B	[28]
**117**	Militarine	B	[27,28]				

**Table 8 molecules-24-03195-t008:** Other compounds from *Pleione* genus.

No.	Compound	Plant	Reference	No.	Compound	Plant	Reference
**125**	5, 7-dihydroxy-8-methoxyflavone	B	[38]	**155**	3-hydroxybenzoic acid	B	[49]
**126**	Isorhamnetin-3, 7-di-*O*-*β*-d-glucopyranoside	B	[29]	**156**	Methyl 3-(3-hydroxyphenyl)Propionate	B	[31]
**127**	3′-*O*-methylquercetin-3-*O*-*β*-d-gluCopyranoside	B	[29]	**157**	4-(4′′-hydroxybenzyl)-3-(3′-hydroxy-phenethyl) furan	B, Y	[49]
**128**	3, 5, 7, 3′-tetrahydroxy-8, 4′-dimethoxy-6-(3-methylbut-2-enyl)flavone	B	[29]	**158**	Methyl 3-(4-hydroxyphenyl)propionate	B	[1]
**129**	3, 5, 3′-trihydroxy-8, 4′-dimethoxy-7-(3-methylbut-2-enyloxy) Flavone	B	[29]	**159**	3-(3′-hydroxyphenethyl)furan-2(5*H*)-one	B	[49]
**130**	Kayaflavone	B	[40]	**160**	Ergosta-4, 6, 8(14), 22-tetraen-3-one	B	[33]
**131**	5, 5′′, 7, 4′, 4′′′, 7′′-hexadroxy-[3′-8′′]Biflavone	B	[40]	**161**	Tetracosanol	Y	[26]
**132**–**133**	Sanjidin A-B	B	[16,40]	**162**	(*E*)-ferulic acid hexacosyl ester	Y	[34]
**134**	Phillygenin	B	[38]	**163**	(*Z*)-ferulic acid hexacosyl ester	Y	[34]
**135**	Epipinoresinol	B	[22]	**164**	Gallicacid	Y	[26]
**136**	Syringaresinol	B, Y	[22,33]	**165**	5-hydroxymethylfurfural	B	[31]
**137**	Syringaresinol Mono-*O*-*β*-d-glucoside	B	[27]	**166**	Methyl 3-(3′-hydroxyphenethyl)furan-2(5*H*)-one	B	[49]
**138**	Lirioresinol	B	[29]	**167**	(24*R*)-cyclomargenyl *p*-coumarate	F	[24]
**139**	(*E*)-*p*-hydroxycinnamic acid	Y	[45]	**168**	(24*R*)-cyclomargeno	F	[24]
**140**	(7*S*, 8*R*)-dehydrodiconiferyl alcohol-9′-*O*-*β*-d-glucopyranoside	B	[27]	**169**	Tetacosanoic acid-2, 3-dihydroxypropyl ester	Y	[26]
**141**	Pleionin	B	[16]	**170**	Hydroquinone	B	[40]
**142**	Pleionol	B	[18]	**171**	*β*-sitosterol	B, Y	[22,31]
**143**	Gastrodioside	B	[29]	**172**	*β*-daucosterol	Y	[34]
**144**	Phenl-*β*-d-glucopyranoside	B	[29]	**173**	Methyl 4-hydroxyphenylacetate	Y	[40]
**145**	Gastrodin	B	[27,45]	**174**	Daucostero	B, Y	[22]
**146**	(*E*)-ferulic acid	Y	[34]	**175**	Physcion	B	[40]
**147**	Cinnamic acid	B	[1]	**176**	Chrysophanol	B	[26]
**148**	*p*-hydroxybenzoic acid	B, Y	[39,40]	**177**	4, 4′-dihydroxydiphenylmethane	B	[38]
**149**	*p*-hydroxybenzaldehyde	B	[39,40]	**178**	4-oxopentanoic	B	[1]
**150**	Methy (4-OH) phenylacetate	B	[25]	**179**	Monopalmttin	Y	[26]
**151**	4-(ethoxymethyl)phenol	B	[1]	**180**	Amber Acid	Y	[22]
**152**	4-(methoxymethyl)phenol	B	[1]	**181**	Adenosine	Y	[22]
**153**	*p*-dihydroxy benzene	B	[35,37]	**182**	Pholidotin	Y	[34]
**154**	3-hydroxybenzenepropanoic acid	B	[45]	**183**	Triphyllol	Y	[34]

**Table 9 molecules-24-03195-t009:** The LPS-stimulated NO production of BV-2 microglail cells.

No.	IC_50_ (μM)	No.	IC_50_ (μM)
**1**	2.35	**82**	21.0
**2**	1.74	**86**	21.8
**63**	2.46	**81**	26.1
**64**	3.14	**14**	14.4
**58**	50.2	**12**	24.5
**62**	38.0	**26**	13.1
**75**	23.2	**10**	17.7
**74**	17.5	**25**	23.4
**73**	56.7	**117**	59.2

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
