# Peer review of "Chemical Constituents and Biological Activity Profiles on Pleione (Orchidaceae)"

_molecules, 2019, doi:10.3390/molecules24173195_

Round 1

Reviewer 1 Report

The manuscript in reference is an up-to-date compilation of the chemistry and biological activity of the genus Pleione (Orchidaceae). The manuscript is interesting and includes important information about this orchid. However, this manuscript suffers from several points that should be addressed prior acceptance.

The manuscript must be scrutinized to correct/revise the problems/mistakes in grammar and style. This is the most important point because it is very difficult to follow some ideas/sentences. An English editing service is strongly recommended. Title: Why is mentioned "pharmacological profile" if only in vitro biological activities are compiled without connection between them? No information regarding pre-clinical or clinical trials are mentioned, so I am not sure if such a term is appropriated. I recommend to change it to "pharmacological or biological activity". Additionally, I recommend to change the title without the word "review" and include "chemical constituents" instead "chemical structures". Lines 17-19: After "In addition," this sentence has no sense and seems to be out of the main idea of the abstract. Lines 19-21: Revise also the meaning of this final sentence because is very confusing. Line 71: Is this a systematic review? Please check that and correct. Line 74: Place "chemical constituents" instead "chemical structures" Line 75: I suggest to add a subsection including information about plant parts and extraction methods as well as spectroscopic data useful for structural elucidation of constituents from Pleione plants. Place bold numbers below structures in figures. Line 196 and forward: I consider a big part of the information about biological activity of Pleione compounds can be better compiled in a table to facilitate to find the information to readers, comprising compound name and number, compound type, source and plant part, biological activity parameter (e.g., IC50), results, observations, etc. Lines 194-195: Revise the meaning of this last sentence. I recommend to organize the conclusions to a more comprehensive way, since after reading, it seems to be unordered. Some ideas are repeated and/or incomplete. For instance, but no limited to, lines 284-285 and line 292 have similar information.

Author Response

Dear reviewer:

    Thank you for your advises and patience. I have revised the manuscript according to your suggestion. The details are listed as following.

Question 1: The manuscript must be scrutinized to correct/revise the problems/mistakes in grammar and style. This is the most important point because it is very difficult to follow some ideas/sentences. An English editing service is strongly recommended.

Answer: Thank you for your advice and patience. I am sorry that I did not use the English editing service before submitting the manuscript. I am so sorry for taking you so much time to read it. The manuscript has been corrected by the English editing service during the revision.

Question 2: I recommend to change it to "pharmacological or biological activity".

Answer: This is a valuable question. I have changed "pharmacological" to "biological activity". (see the title)

Question 3: I recommend to change the title without the word "review".

Answer: Thank you for your kind advice. I have changed the title without the word "review". (see the title)

Question 4: "chemical constituents" instead "chemical structures"

Answer: Thank you for your suggestion. I have used "chemical constituents" instead of "chemical structures".(see the Line 73)

Question 5: Lines 17-19: After "In addition," this sentence has no sense and seems to be out of the main idea of the abstract.

Answer: Thank you for your question. But in abstract, I introduced the types of the compounds, and then I want to introduce the biologic activities about them. I think it is reasonable and necessary in this article.

Question 6: Lines 19-21: Revise also the meaning of this final sentence because is very confusing.

Answer: Thank you for your advice. I have revised the sentence. (see Line19-20)

Question 7: Is this a systematic review? Please check that and correct.

Answer: Thank for your rigorous review. The article just refers to the chemical aspect. It is not really a systematic study.(see Line 71-72)

Question 8: Line 75: I suggest to add a subsection including information about plant parts and extraction methods as well as spectroscopic data useful for structural elucidation of constituents from Pleione plants.

Answer: Thank you for your valuable suggestion. I think it is not representative if I just listed spectroscopic data of some compounds, and could not conclude some laws well. So I chose to list the spectroscopic data of core structures of all the phenanthrenes and bibenzyls, even though it was a large amount of work. The plant parts are pseudobulbs of all the chemical research mentioned in the manuscript, so I did not show the information in the table.

Question 9: Place bold numbers below structures in figures.

Answer: Thank you for your advice. I have placed all the numbers below structures in figures.

Question 10: Line 196 and forward: I consider a big part of the information about biological activity of Pleione compounds can be better compiled in a table to facilitate to find the information to readers, comprising compound name and number, compound type, source and plant part, biological activity parameter (e.g., IC50), results, observations, etc.

Answer: Thank you for your suggestion. I tried to make a table according to your valuable advice in the beginning. But it turned out that the table looks not scientific and may not be pretty. Because the description is variety, such as IC50, cell survival rate and cell inhibition rate, and even just describe by some words. So I finally decided to make a table just about the NO production of BV-2 microglail cells.

Question 11: Lines 194-195: Revise the meaning of this last sentence.

Answer: Thank you for your advice. I have revised this sentence. (see the Line 202-203)

Question 12:I recommend to organize the conclusions to a more comprehensive way, since after reading, it seems to be unordered. Some ideas are repeated and/or incomplete. For instance, but no limited to, lines 284-285 and line 292 have similar information.

Answer: Thank you for your advice. Indeed, some ideas were repeated and it created additional difficulties for the reader. I have re-organized the conclusions and made it more comprehensive.

Reviewer 2 Report

The manuscript submitted is the review on chemical structures and pharmacological activity of secondary metabolites present in selected Pleione species.

Although the authors put a huge effort into collecting data published over the past two decades (1996-2019), the language used for their presentation is full of mistakes. Sentences used have a bad grammatical construction, sometimes they lack verbs or nouns or even subjects, what makes submitted manuscript hard to read.

Line 62: what is DHHP radical scavenging assay? I cannot find any literature reference cited in the text relating to this issue.

I don’t see the reason to put all those lines-long-chemical-names of structures in the text when each compound has its own number and individual formulas are included in the tables and on figures. This only creates additional difficulties for the reader, since proper content of sentences is lost in the thicket of chemical names. 

I strongly advise a detailed edition of the text in terms of the language used before any further resubmission.

Author Response

Dear reviewer:

     Thank you for your advises and patience. I have revised the manuscript according to your suggestion. The details are listed as following.

Question 1: The language used for their presentation is full of mistakes. Sentences used have a bad grammatical construction, sometimes they lack verbs or nouns or even subjects, what makes submitted manuscript hard to read.

Answer: Thank you for your advice and patience. I am sorry that I did not use the English editing service before submitting the manuscript. I am so sorry for taking you so much time to read. The manuscript has been corrected by the English editing service.

Question 2: what is DHHP radical scavenging assay? I cannot find any literature reference cited in the text relating to this issue.

Answer: Thank you for your question. The DHHP is one of the radical scavenging assay. I found it was used rarely in research. So I used "scavenging assay" instead of DHHP radical scavenging assay. (see the Line 61)

Question 3: I don’t see the reason to put all those lines-long-chemical-names of structures in the text when each compound has its own number and individual formulas are included in the tables and on figures. This only creates additional difficulties for the reader, since proper content of sentences is lost in the thicket of chemical names.

Answer: Thank you for your valuable suggestion. I have just shown the number of compounds in the text.

Round 2

Reviewer 1 Report

Authors adequately addressed my comments and suggestion. The manuscript looks better and improved in quality and content. I consider this manuscript can be accepted in the current form.

Author Response

Dear reviewer:

  Thank you for your patience and affirmation of the manuscript. Due to your suggestion, the content of the manuscript has improved a lot. I have learned a lot from this submission experience the most import thing is to improve my English.

Kind regard

Wu xiaoqian

Reviewer 2 Report

Dear Authors,

thank you for your responses. I see a significant improvement after reading the modified version of the manuscript.

Question 2: what is DHHP radical scavenging assay? I cannot find any literature reference cited in the text relating to this issue.

Answer: Thank you for your question. The DHHP is one of the radical scavenging assay. I found it was used rarely in research. So I used "scavenging assay" instead of DHHP radical scavenging assay. (see the Line 61) 

If I understand correctly it should be then DPPH, not DHHP, right?

Thank you for organizing all compounds by numbers and for putting them into Tables.

Please correct the References because plant species in titles cited are not always written in italic.

Author Response

Dear reviewer:

    I am grateful for your suggestion. It was my careless that I wrote the wrong DPPH.

    Thank you for your patience and affirmation of the manuscript. Due to your suggestion, the content of the manuscript has improved a lot. I have learned a lot from this submission experience the most import thing is to improve my English.

Kind regard

Wu xiaoqian